# Tautomerism unveils a self-inhibition mechanism of crystallization

Weiwei Tang [1,2], Taimin Yang [3], Cristian A. Morales-Rivera[4], Xi Geng[1], Vijay K. Srirambhatla[5,6], Xiang Kang[2], Vraj P. Chauhan[1], Sungil Hong[4], Qing Tu[7], Alastair J. Florence [5,6], Huaping Mo[8], Hector A. Calderon[9,10], Christian Kisielowski [10], Francisco C. Robles Hernandez [11], Xiaodong Zou [3], Giannis Mpourmpakis[4] & Jeffrey D. Rimer [1] ✉

Modifiers are commonly used in natural, biological, and synthetic crystallization to tailor the growth of diverse materials. Here, we identify tautomers as a new class of modifiers where the dynamic interconversion between solute and its corresponding tautomer(s) produces native crystal growth inhibitors. The macroscopic and microscopic effects imposed by inhibitor-crystal interactions reveal dual mechanisms of inhibition where tautomer occlusion within crystals that leads to natural bending, tunes elastic modulus, and selectively alters the rate of crystal dissolution. Our study focuses on ammonium urate crystallization and shows that the keto-enol form of urate, which exists as a minor tautomer, is a potent inhibitor that nearly suppresses crystal growth at select solution alkalinity and supersaturation. The generalizability of this phenomenon is demonstrated for two additional tautomers with relevance to biological systems and pharmaceuticals. These findings offer potential routes in crystal engineering to strategically control the mechanical or physicochemical properties of tautomeric materials.

Modifiers of diverse materials[1–6] exhibit structures and/or compositions that differ from a solute molecule but often contain similar functional motifs that facilitate molecular recognition for modifier binding to crystal surfaces[7–9]. Here we examine the intrinsic capability of tautomers[10–12], or structural isomers, to operate as crystal growth inhibitors. Molecular tautomers are prevalent in many natural and synthetic crystals[13–15] where they play a central role in the development of pharmaceutical and biological drugs[12] and materials with diverse

optical[16], magnetic[17], electrical[18], and chemical[19] properties. Tautomerism is a ubiquitous phenomenon in nature widely presented in many small organics[12] and bio-macromolecules[11] such as peptides, nucleic acids, and proteins. These naturally forming structural isomers undergo interconversions via intramolecular relocation of atoms, most commonly hydrogens (i.e., prototropic tautomerism[20]). Tautomer interconversion rates in solution are less rapid than conformers, slightly faster than those of diastereomers (e.g., epimers, anomers),

[1]University of Houston, Chemical and Biomolecular Engineering, Houston, TX 77204, USA. [2]Tianjin University, School of Chemical Engineering and Technology, State Key Laboratory of Chemical Engineering, The Co-Innovation Center of Chemistry and Chemical Engineering of Tianjin, Tianjin 300072, China. [3]Stockholm University, Department of Materials and Environmental Chemistry, SE-106 91 Stockholm, Sweden. [4]University of Pittsburgh, Chemical and Petroleum Engineering, Pittsburgh, PA 15261, USA. [5]EPSRC Future Manufacturing Research Hub for Continuous and Manufacturing and Advanced Crystallization (CMAC), University of Strathclyde, Technology and Innovation Centre, 99 George Street, Glasgow G1 1RD Scotland, UK. [6]Strathclyde Institute of Pharmacy and Biomedical Sciences, University of Strathclyde, Glasgow G4 0RE Scotland, UK. [7]Texas A&M University, Materials Science & Engineering, College Station, TX 77843, USA. [8]Purdue University, Medicinal Chemistry and Molecular Pharmacology, College of Pharmacy, West Lafayette, IN 47097, USA. [9]Instituto Politecnico Nacional, ESFM-IPN, Departamento de Física, UPALM Zacatenco, Mexico City CDMX 07338, Mexico. [10]The Molecular Foundry, Lawrence Berkeley National Laboratory, One Cyclotron Rd., Berkeley, CA 94720, USA. [11]University of Houston, Mechanical Engineering Technology, Houston, TX 77204, USA. ✉e-mail: jrimer@central.uh.edu

and markedly faster than functional, skeletal, cis-/trans-, enantiomer, or positional isomers[10,21]. Crystallization of molecules with multiple conformers can lead to polymorphism[22], while stereoisomers[1] and anomers[23,24] can function as crystal growth modifiers that alter crystallization rates and/or morphology. Studies of growth kinetics[25,26] indicate that the presence of multiple conformers in solution generally does not impact crystal growth[27] owing to the rapid rate of conformer exchange prior to solute incorporation into the crystal lattice. Several studies have posited[28,29] that energetic barriers for conformational exchange (ca. 30–40 kJ/mol) could impact crystallization, but evidence for this effect has been elusive. In the case of tautomers, the exchange between structural isomers typically involves energy barriers greater than 40 kJ/mol where the relatively slow rate of

interconversion[10] is comparable to characteristic timescales of solute incorporation into crystals[30]. Herein we investigate this effect on the crystallization of ammonium urate ($NH_4HU$), a prototypical tautomeric compound that is a pathological component of kidney stones[31,32].

## Results

### Crystal structure determination and macroscopic crystallization assays

We first prepared $NH_4HU$ crystals in an alkaline ammonium urate growth solution (pH 11) as a reference condition (control) for crystallization studies and for the purpose of $NH_4HU$ structure determination. In basic media, $NH_4HU$ crystallizes as nanorods (Fig. 1a, b). Three-dimensional electron diffraction (3D ED) was performed at low

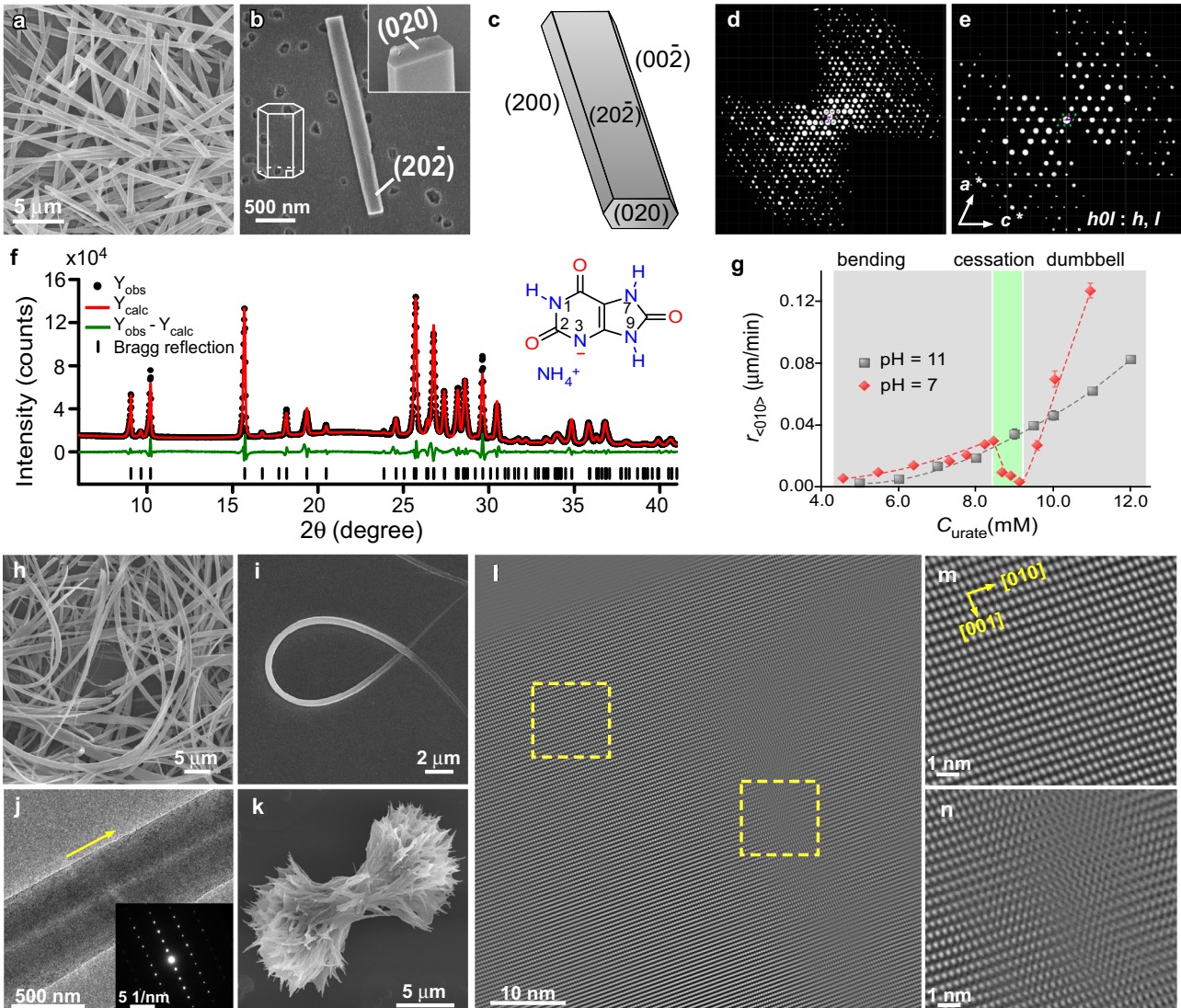

**Fig. 1 | Structure and morphology of $NH_4HU$ crystals. a, b** SEM images of rod-like $NH_4HU$ crystals prepared at pH 11 and 12 mM urate. The inset of panel **b** shows the end face of crystals with a hexagonal cross-section. **c** Schematic diagram of indexed $NH_4HU$ crystal facets. **d** Typical 3D reconstructed reciprocal lattice of a single $NH_4HU$ crystal viewed along the *b*-axis. The crystal facets could be indexed from the 3D ED data, as shown in panel **c**. **e** 2D (*h0l*) slice cut from the reconstructed 3D reciprocal lattice of $NH_4HU$ showing reflection condition *h0l*: $h = 2n$ and $l = 2n$. **f** Pawley profile fitting (red line) of the powder XRD pattern (black dots) with difference plot (green line) of $NH_4HU$ crystals. The black tick marks at the bottom represent the positions of Bragg reflections. Inset: Molecular structure of $NH_4HU$ showing ionization at the nitrogen (N3) site. **g** Microfluidics measurements of macroscopic growth rates along the *b*-direction, $r_{<010>}$, at pH 11 (control, gray

squares) and pH 7 (red diamonds) as a function of urate concentration. Data are the average measurements of 10 to 15 crystals and error bars span two standard deviations. Dashed lines are interpolated to serve as a guide to the eye. The gray- and green-shaded regions distinguish growth regimes in pH 7 solutions. **h, i** SEM images of $NH_4HU$ crystals prepared at pH 7 and 6–7 mM urate. **j** TEM image showing the straight segment of a bent $NH_4HU$ crystal. Inset: Corresponding SAED pattern revealing the growth orientation along the <010> direction. Scale bar equals 5 nm⁻¹. **k** SEM image of a crystal prepared at pH 7 and 14 mM urate with a dumbbell-shaped morphology. **l** Filtered TEM image as determined by a low-dose (4 e⁻/Å²) fractionation technique showing the atomic clusters of the structure (dots in the image) with enlarged images of dashed boxes in panel l showing the absence (**m**) and presence (**n**) of defects (see Supplementary Fig. 6).

electron dosage and collected at different tilt angles. The 3D reciprocal lattice of $NH_4HU$ crystal reconstructed from the 3D ED data (Fig. 1d, e, Supplementary Fig. 1a–c, and Supplementary Movie 1) shows that the crystal has a monoclinic space group with $a = 21.385(4)$ Å, $b = 3.5300(7)$ Å, $c = 20.080(4)$ Å, and $\beta = 114.10(3)°$. The short $b$-axis is along the rod direction and a cross-section (Fig. 1b, inset) with facets is indexed in Fig. 1c. The $NH_4HU$ crystal diffracted to a very high resolution (0.85 Å) and its structure was solved from the 3D ED data in space group $C2/c$ (Supplementary Table 1). Direct determination of hydrogen positions made it possible to unambiguously identify the correct tautomer species in the crystal, as shown as an inset in Fig. 1f (see also Supplementary Fig. 2). The structure was also refined by powder X-ray diffraction (XRD) data using the Rietveld method (Supplementary Fig. 1d) to confirm the space group. The Pawley fitting profile (Fig. 1f) provided final refined lattice parameters (see Supplementary Table 1 and Supplementary Section 1.3). The asymmetric unit contains equimolar amounts of $NH_4^+$ and $HU^-$ (Fig. 1f, inset), which was confirmed by carbon, hydrogen, and nitrogen elemental analysis (Supplementary Table 2). These collective analyses provide the first accurate determination of the ammonium urate crystal structure, which differs from those previously proposed in literature[33] (Supplementary Fig. 2), positing different molecular formula.

Uric acid is a polyprotic acid with two dissociation constants ($pKa = 5.4$ and $10.3$; Supplementary Fig. 3a)[32,34]. The divalent form is the dominant species in highly alkaline media, but 17% of the total urate concentration at pH 11 is monovalent urate $HU^-$ (Supplementary Fig. 3a), the principal component of $NH_4HU$ crystals. The kinetics of bulk $NH_4HU$ growth was measured in situ using microfluidics[35] to monitor temporal changes in macroscopic crystal dimension along the $b$-direction (i.e., length of the rod) at room temperature. The rate of growth, $r_{<010>}$, was measured in an alkaline medium (pH 11) as a function of the total concentration of urate species, $C_{urate}$, using a continuous flow rate of supersaturated growth solution (see Methods). The resulting super-linear increase in $r_{<010>}$ with increasing $C_{urate}$ (Fig. 1g, gray symbols) is typical of classical crystallization[30,36] where growth occurs by layer generation and spreading (Supplementary Fig. 4d). When the same measurements were performed in a growth medium at neutral conditions (pH 7), where the monovalent form of urate is the most dominant species, there is a deviation in the trend of $r_{<010>}$ (Fig. 1g, red symbols). At intermediate supersaturation, the rate of growth suddenly decreases to the point of near complete cessation, followed by a rapid increase in crystallization wherein the rate of growth at higher $C_{urate}$ is approximately linear and much faster than $r_{<010>}$ in the alkaline growth solution. The range of urate concentration where growth cessation occurs is relatively narrow ($8.5 < C_{urate} < 9.5$ mM). At urate concentrations below this range, scanning electron microscopy (SEM) images of $NH_4HU$ crystals extracted from bulk crystallization assays at quiescent conditions reveal natural bending (Fig. 1h), leading to a spaghetti-like morphology (Fig. 1i). Selected area electron diffraction (SAED) patterns (Fig. 1j) and powder XRD (Supplementary Fig. 5) confirm the crystallinity of bent $NH_4HU$. At concentrations above growth cessation, we observe dendritic growth leading to a dumbbell shape (Fig. 1k) that can evolve into a spherulite-like morphology (Supplementary Fig. 3b). These hierarchical structures mimic the features of crystals extracted from pathological stones[32,37], where physiological pH 5–8 matches that of our in vitro experiments. Transmission electron microscope (TEM) images of straight segments in bent crystals (Fig. 1l) and magnified regions (Fig. 1m, n) exhibit remarkable resolution of $NH_4HU$ crystals showing the atomic clusters of the organic crystals as dots. The cluster arrangements clearly show crystalline structural ordering along with the presence of defects (vide infra). The Fourier Transform Filtered images in Fig. 1l–n were obtained by an electron microscope technique that combines the use of rather low doses to preserve the genuine

structure of the sample with a limitation on the number of images (see Supplementary Fig. 6 for unfiltered images).

## Microscopic analysis of crystal growth and self-inhibition dynamics

Microscopic evidence of different $NH_4HU$ crystal growth regimes was extracted from in situ atomic force microscopy (AFM) measurements on basal ($20\bar{2}$) crystal surfaces as a function of growth solution alkalinity and urate concentration. Working with a supersaturated growth solution at pH 7 in the regime corresponding to crystal bending ($C_{urate} = 8.1$ mM), we observe the birth and spreading of two-dimensional (2D) islands. Time-resolved images from Supplementary Movie 2 show nuclei (islands I–III in Fig. 2a) that continue to grow with increased imaging time, and a smaller population of islands (IV in Fig. 2a) with radii $R$ that dissolve. The critical radius of 2D nucleation, $R_c$, is estimated from sequential AFM images by determining a threshold radius above which islands have a higher probability to grow ($R > R_c$), and below which islands are more likely to dissolve ($R < R_c$)[6,38]. During the growth of new layers, the step velocity in the <010> direction, $v_{<010>}$, is much faster than that in the <101> direction, leading to an anisotropic morphology that mimics the bulk crystal habit. Measurements of step velocity as a function of interstep distance (Supplementary Fig. 7b) reveal a signature profile[39] indicative of solute incorporation via a surface diffusion pathway. An unusual observation is the oriented growth of islands wherein steps advance in a direction [uvw] (Fig. 2a, yellow arrow) at angles [uvw]∩[010] between −25° and +25°. The distribution of [uvw]∩[101] angles for 2D islands, labeled $\theta_1$ in Fig. 2a, has an average value of 81° (Fig. 2b). We also observe highly corrugated steps with large protrusions (dashed circles) that advance at different orientations with a narrower distribution of [uvw]∩[101] angles, labeled $\theta_2$ in Fig. 2a, and an average value of 86° (Fig. 2b). The skewed angles of layer advancement at a microscopic level are consistent with macroscopic angles observed in tapered $NH_4HU$ crystals (vide infra).

AFM measurements of step velocity $v_{<010>}$ as a function of increasing urate concentration (Fig. 2c) reveal an identical trend to macroscopic growth kinetics measured by microfluidics (Fig. 1g), including similar values of $C_{urate}$ corresponding to the sudden drop in step velocity in the regime of growth cessation. In situ AFM measurements were also performed at higher urate concentrations to assess differences in layer growth within the regime where crystal branching leads to dumbbell formation. For these studies, we first stabilized the crystal substrate at a urate concentration (4 mM) slightly above $NH_4HU$ solubility (2.8 mM)[32] where we observe a "dead zone" ($v_{<010>} = 0$ nm/s; orange region in Fig. 2c). The latter is consistent with a reported phenomenon in literature where crystal growth modifiers (or impurities) are present in low concentrations[30,40], while other factors such as solvent interactions and molecular-scale surface roughness may contribute to the presence of a dead zone[41,42]. Once the urate concentration is increased to 12 mM, within minutes, we observe the rapid nucleation and growth of highly anisotropic layers (Supplementary Movie 3) comprised of step bunches where unidirectional advancement occurs in either the [010] direction (Fig. 2d, yellow arrow) or [0$\bar{1}$0] direction (Fig. 2d, white arrow). Time-resolved AFM measurements were conducted at 10 mM urate, where the evolution of step bunch height and rate of advancement could be accurately monitored (Supplementary Movie 4). As shown in Fig. 2e, a step bunch with length 1→2 advances in the [0$\bar{1}$0] direction. Height profiles at periodic imaging times (Fig. 2f) show that step bunch growth is three-dimensional. The initial step bunch height is ca. 2 nm, which is nearly twice the length of a urate molecule (~0.9 nm). As the step bunch advances unilaterally in length (from 50 to 130 nm over 526 s), the height of the advancing step front increases to 5 nm while the opposite end remains fixed at a height equal to that of the original step bunch.

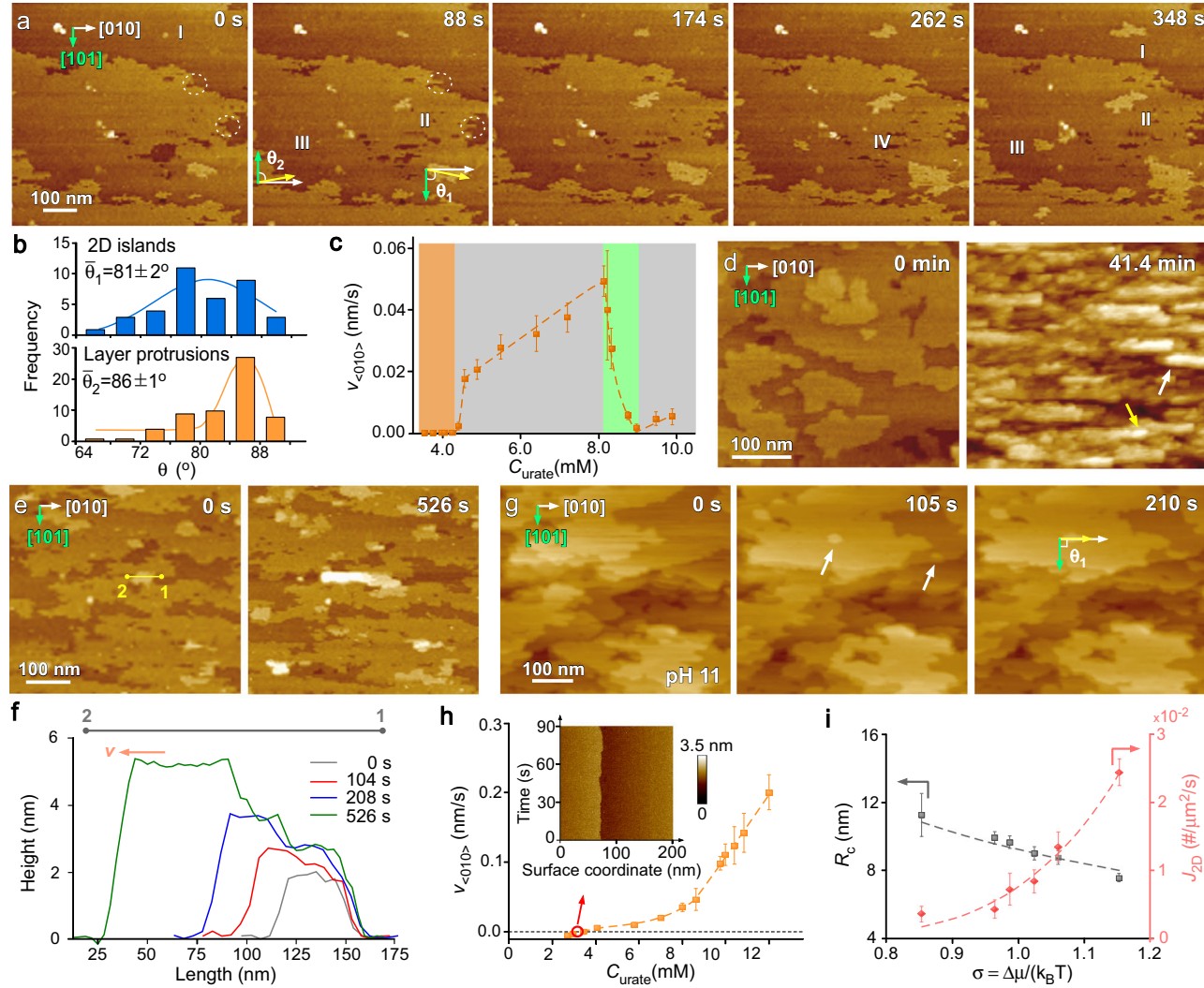

**Fig. 2 | Time-resolved AFM images of NH₄HU surface growth. a** Tapping mode AFM images of (20$\bar{2}$) surface growth in situ (pH 7 and 8.1 mM urate) extracted from Supplementary Movie 2 show both growing (I–III) and dissolving (IV) islands, and corrugated steps with protrusions (dashed circles). **b** Orientation angles [uvw]∩[101] for 2D island ($\theta_1$, blue) and step protrusion ($\theta_2$, orange) advancement ($n \geq 40$ measurements of the same sample. **c** Step velocity of layer advancement at pH 7 as a function of urate concentration. The shaded regions indicate a "dead zone" (orange), step growth (gray), and growth cessation (green). Symbols are the average of 10 to 15 measurements and error bars span two standard deviations. **d** Time-elapsed images from Supplementary Movie 3 showing the evolution of multilayer step bunches in a growth solution at pH 7 and 12 mM urate ($\sigma = 1.30$). Yellow and white arrows highlight step bunch advancement in the +b and -b directions, respectively. **e** AFM images from Supplementary Movie 4 showing the onset of step bunching in a growth solution at pH 7 and 9.5 mM urate ($\sigma = 1.09$).

**f** Height profiles of line $l_{1\text{-}2}$ in panel e at various time intervals where a single step is 0.9 nm in height. **g** Time-elapsed images from Supplementary Movie 5 showing 2D island generation and spreading in a control growth solution (pH 11) with 12 mM urate ($\sigma = 1.15$). White arrows (at 105 s) highlight nuclei that advance in the *b* direction at angle $\theta_1 = 90°$. **h** Step velocity at pH 11 where symbols are the average of 10 to 15 measurements and error bars span two standard deviations. Inset: In situ AFM kymograph collected with disabled scanning along the y-axis. **i** Dependence of the critical radius of 2D island nucleation $R_c$ and the rate of island nucleation $J_{2D}$ on the crystallization driving force $\sigma = \Delta\mu/k_BT$ at pH 11. Symbols are the average of 25 to 40 measurements and error bars span two standard deviations. The gray dashed line is a plot of the Gibbs–Thomson relation $R_c = \Omega\alpha/\Delta\mu$ with step line tension $\alpha = 247 \pm 4$ mJ m$^{-2}$. The red dashed line is an exponential fit ($J_{2D} \propto \exp(-\Delta G_{2D}^*/RT)$) where $\Delta G_{2D}^*$ is the energy barrier of 2D layer nucleation.

AFM measurements of NH₄HU surface growth in alkaline media (pH 11) reveal 2D layer generation and spreading without the abnormalities observed at lower pH. Time-resolved images for 12 mM urate (Supplementary Movie 5) show island nucleation (Fig. 2g, white arrows) where growth occurs preferentially in the <010> direction (i.e., $\theta_1 = 90°$) and unfinished layers exhibit corrugated edges without noticeable step bunching. The increase in $v_{<010>}$ with increasing urate concentration (Fig. 2h) exhibits a trend that closely aligns with reported cases of classical crystallization in literature[30,40] without growth cessation observed at neutral conditions. The solubility of NH₄HU crystals shifts to a slightly higher value (3.4 mM, Supplementary Fig. 4c) in alkaline media, which was confirmed by in situ AFM imaging with disabled scanning along the y-axis where the resulting

kymograph (Fig. 2h, inset) shows a step edge in equilibrium with the solution (i.e. the step neither advances nor recedes with scanning time). Sequential AFM images at each urate concentration were analyzed according to a reported procedure[6] to obtain the critical radius of 2D island nucleation $R_c$ and the rate of island nucleation $J_{2D}$ (Fig. 2i), which both exhibit trends with increasing supersaturation ($0.85 < \sigma < 1.15$) that are consistent with classical models[43].

## Confirmation of urate tautomers in growth media

Several observations of NH₄HU crystallization at pH 7 are indicative of growth in the presence of an inhibitor. These include multi-oriented step advancement, the increased density of corrugated step edges, the generation of step bunches, evidence of a dead zone, and the unusual

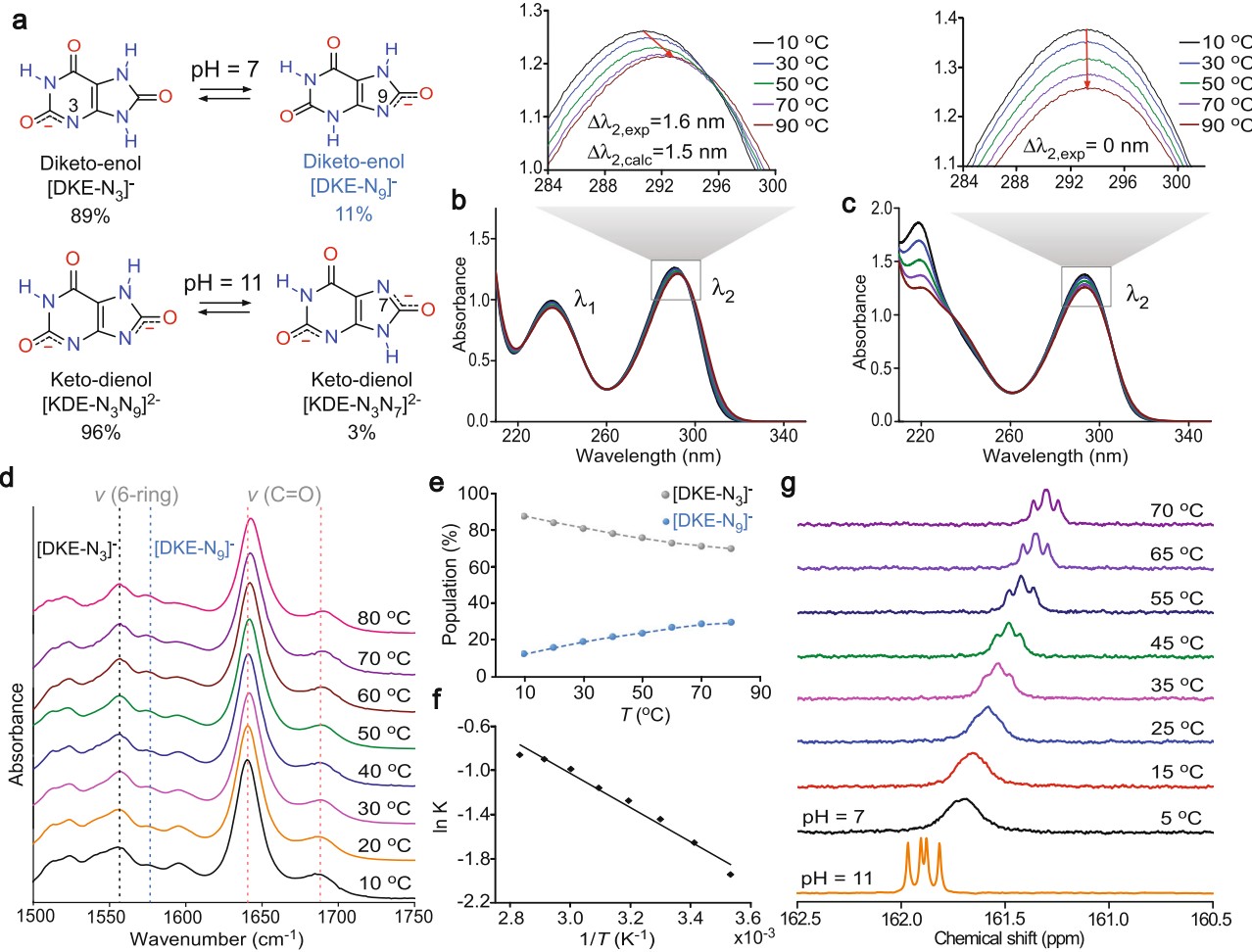

**Fig. 3 | Experimental and computational evidence of urate tautomers.**
**a** Chemical equilibrium between urate diketo-enol tautomers at pH 7 and keto-dienol tautomers at pH 11 (control). Percentages of tautomers are based on first-principles (CBS-QB3) calculations at 25 °C on mono and double deprotonated urate tautomers simulating species at pH 7 and pH 11, respectively. **b** Variable-temperature UV-Vis spectra of an aqueous 1 mM urate solution at pH 7. Inset: Bathochromic shift of the absorption peak $\lambda_2$ with increasing temperature with maximum peak shifts from the experiment ($\Delta\lambda_{2,exp}$) and DFT ($\Delta\lambda_{2,calc}$, Supplementary Fig. 10c). **c** Variable-temperature UV-Vis spectra of a 1 mM urate solution at pH 11. Inset: Absorption peak $\lambda_2$ with increasing temperature shows no shift. **d** FTIR spectra of 18 mM urate in $D_2O$ at different temperatures. The asymmetric stretching modes of the six-membered ring show increasing intensity of peaks at 1576 cm$^{-1}$ ([DKE-N$_9$]$^-$) relative to 1556 cm$^{-1}$ ([DKE-N$_3$]$^-$) with increasing temperature. There is a

concomitant red shift of carbonyl group asymmetric vibration bands (see Supplementary Fig. 11a for IR peak assignments). **e** Population of [DKE-N$_9$]$^-$ and [DKE-N$_3$]$^-$ tautomers derived from the relative intensity of IR absorption peaks at 1576 and 1556 cm$^{-1}$. **f** Temperature dependence of the equilibrium constant $K$ at pH 7 between tautomers [DKE-N$_9$]$^-$ and [DKE-N$_3$]$^-$ determined from the relative intensity of IR peaks at 1556 and 1576 cm$^{-1}$. **g** Variable-temperature $^{13}$C-NMR spectra of 10 mM urate (2–$^{13}$C,1,3,7–$^{15}$N3 atomic labeled) in 95% (v/v) H$_2$O/D$_2$O at pH 7 comparison with the same concentration at pH 11 (bottom). In neutral solutions, significant peak broadening transitions to gradual peak sharpening with increasing temperature, reflecting the nature of interconversion between tautomers [DKE-N$_9$]$^-$ and [DKE-N$_3$]$^-$ (Supplementary Fig. 7a and Supplementary Movie 6). The appearance of a doublet-doublet peak is due to the scalar couplings from two adjacent, labeled $^{15}$N resonances.

appearance of growth cessation in a narrow range of urate concentration. We hypothesize that all these observations can be correlated to urate tautomerization, wherein the minor tautomer, which is non-native to the crystal structure, functions as a crystal growth inhibitor. Here we use a combination of spectroscopy measurements and first-principles calculations to examine the stability of tautomers in aqueous solutions at pH 7 and 11. Free energy CBS-QB3 calculations at pH 7 and room temperature, including implicit water solvation, predict out of 30 total urate tautomers (Supplementary Fig. 8), only two thermodynamically favorable diketo-enol structures (Fig. 3a): [DKE-N$_3$]$^-$ and [DKE-N$_9$]$^-$. Based on Boltzmann statistics, these tautomers account for 89 and 11% of the urate population, respectively. Similar calculations at pH 11 reveal one predominant structure, keto-dienol [KDE-N$_3$N$_9$]$^{2-}$, and a minor tautomer [KDE-N$_3$N$_7$]$^{2-}$ (Fig. 3a) that account for 96 and 3% of the total urate population, respectively. Nineteen additional structures (Supplementary Fig. 9) collectively account for the remaining 1%.

Boltzmann distributions predict the percentage of minor tautomer at neutral pH increases with temperature; therefore, experimental validation of urate tautomerism was conducted over a range of temperatures between 10 and 90 °C using three separate spectroscopy techniques. UV-Vis spectra at pH 7 (Fig. 3b) exhibit a peak shift from 290 to 291.6 nm ($\Delta\lambda_{2,\,exp} = 1.6$ nm) with increasing temperature, which is nearly equal to the peak shift, $\Delta\lambda_{2,\,calc} = 1.5$ nm, predicted by Density Functional Theory (DFT) calculations (Supplementary Fig. 10c). The UV-Vis spectra at pH 11 (Fig. 3c) display no apparent peak shift owing to the small percentage of minor tautomer [KDE-N$_3$N$_7$]$^{2-}$. Similar trends are observed in Fourier transform infrared (FTIR) spectra at pH 7 (Fig. 3d), where two signature peaks associated with the carbonyl group around 1640 and 1689 cm$^{-1}$ experience a red shift with increasing temperature. There are also two characteristic peaks associated with asymmetric stretching modes of the 6-membered ring at 1556 and 1576 cm$^{-1}$ for [DKE-N$_3$]$^-$ and [DKE-N$_9$]$^-$, respectively, that exhibit changes in relative intensity with increasing temperature.

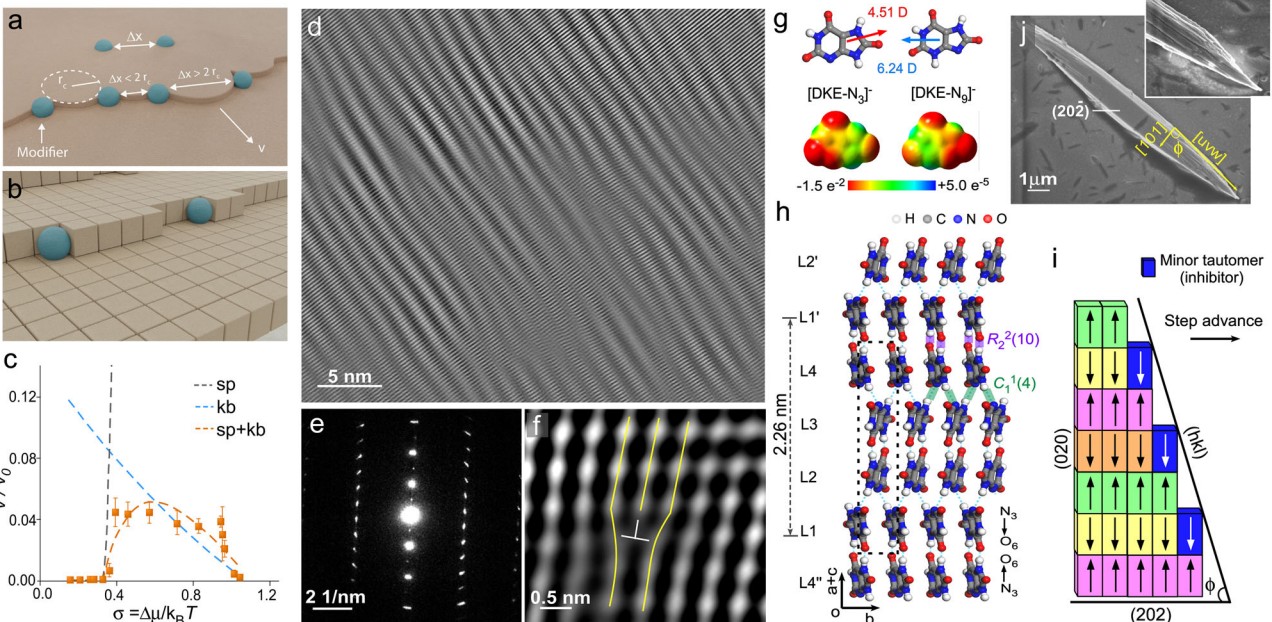

**Fig. 4 | Self-inhibition mechanism of the minor tautomer. a, b** Illustrations of step pinning (**a**) and kink blocking (**b**) mechanisms of growth inhibition[6]. **c** Step velocity $v$ at pH 7 (from Fig. 2c) normalized by $v_0$ measurements in pH 11 corresponding to growth conditions (control) containing a trace quantity of minor tautomer. Model fits are indicated by dashed lines for step pinning (black), kink blocking (blue), and combined mechanisms (orange). See Supplementary Section 2 for details. **d** TEM image of a bent crystal prepared at pH 7 (see also Supplementary Fig. 6). **e** SAED pattern of a bent crystal showing line broadening. **f** Fourier-filtered high-resolution TEM image showing an edge dislocation. The image is produced by combining ten low-dose experimental images and Fourier filtering to show atomic clusters of the organic crystal. **g** Electrostatic potential of tautomers [DKE-N$_3$]$^-$ and [DKE-N$_9$]$^-$ from DFT calculations showing the difference in charge distribution and dipole moments (arrows with 4.51 and 6.24 Debye, respectively). **h** Molecular packing in the NH$_4$HU crystal structure viewed on the (20$\bar{2}$) plane. The unit

cell (dashed box) contains four distinct layers (L1 to L4) differing by their orientation (arrows). Key intermolecular interactions include strong amide-amide hydrogen bonds (purple, $R_2^2(10)$, $d_{N\cdots O} = 2.670\,Å$, $\theta_{NH\cdots O} = 159.13°$) forming urate dimer and amine-nitrogen hydrogen-bonding chains (green, $C_1^1(4)$, $d_{N\cdots N} = 2.944\,Å$, $\theta_{NH\cdots N} = 165.86°$) connecting urate dimers. Ammonium ions are omitted for clarity. **i** Idealized binding of minor tautomer [DKE-N$_9$]$^-$ to (020) step edge sites via $R_2^2(10)$ hydrogen-bonding interactions. Tautomer binding to only L1 or L3 kink sites would produce a vicinal surface with angles ($\phi = 73 - 90°$, Supplementary Fig. 12f) that are consistent with surface layer angles $\theta_2$ in AFM images (Fig. 2b) and tapering angles $\phi$ in SEM images. **j** SEM image of a tapered NH$_4$HU crystal prepared in pH 7 growth solution with 6 mM urate ($\sigma = 1.0$). Inset: Enlarged image of a tapered crystal end where angles $\phi = [uvw] \cap [101]$ span from 72 to 90° (Supplementary Fig. 12c). Measurements of 50 crystals yield an average $\phi = 86 \pm 1°$.

The ratio of FTIR peak intensities for the 6-rings was used to extract the fraction of diketo-enol tautomers as a function of temperature (Fig. 3e) and the corresponding equilibrium constant $K$ (Fig. 3f), which exhibits the expected exponential dependence on temperature, $K \propto \exp(-\Delta H/RT)$[44]. The estimated enthalpy of urate tautomerism $\Delta H$ (12.9 ± 0.8 kJ mol$^{-1}$) is within the range of reported values ($\Delta H = 7-26$ kJ mol$^{-1}$) for other organic tautomers in literature[45].

The existence of urate tautomers was also confirmed by variable-temperature solution $^{13}$C-NMR spectroscopy. As shown in Fig. 3g, the $^{13}$C-NMR spectrum at pH 11 contains a sharp doublet-doublet peak suggesting the absence of the minor tautomer in contrast to the broad peak observed at pH 7 owing to the presence of [DKE-N$_9$]$^-$. With an increasing temperature of the neutral solution, there is a shift from a single broad peak to a doublet-doublet peak that retains a small degree of peak broadening. These measurements reflect the kinetics of interconversion between [DKE-N$_3$]$^-$ and [DKE-N$_9$]$^-$ tautomers, in agreement with DFT calculations indicating an activation barrier of 73.3 kJ mol$^{-1}$ at 25 °C (Supplementary Fig. 7a and Supplementary Movie 6). This indicates a major–minor tautomer exchange rate on the order of milliseconds (Supplementary Table 3), which is within the NMR detection limit, thus explaining why peak broadening is more pronounced at a lower temperature. Collectively, spectroscopy experiments and DFT calculations confirm that growth solutions at pH 11 contain a single urate tautomer, whereas solutions at pH 7 contain a major tautomer with a structure identical to that of urate in the NH$_4$HU crystal structure and a minor tautomer that we posit is a native crystal growth inhibitor.

## Mechanism of self-inhibition by the minor tautomer

There are two common mechanisms of crystal growth inhibition: step pinning (Fig. 4a) and kink blocking (Fig. 4b). Step pinning occurs when modifiers adsorb on terraces or step edges and impose a surface tension on the advancing layer when two adsorbed species are separated by a distance $\Delta x$ smaller than the critical radius of curvature $R_c$ for the step. Kink blocking involves modifier adsorption to kink sites, which are the most favorable for solute incorporation. Neither inhibitor model (Fig. 4c) alone can capture the step velocity profile at neutral pH measured by AFM; however, the experimental data can be fitted using a combination of both mechanisms (Fig. 4c, dashed orange line), indicating the inhibitor [DKE-N$_9$]$^-$ operates by a dual mode of action. This behavior deviates from reported cases involving a binary combination of inhibitors where each acting by a different mechanism results in antagonistic cooperativity[6]. Here the dual effects of the single inhibitor are synergistic, which suggests kink blockers do not significantly lower the line tension of advancing layers. Slight differences between the experiment and model fit (Fig. 4c) suggest additional factors contribute to growth cessation, such as strain imposed by minor tautomer [DKE-N$_9$]$^-$ occlusion within the crystal that is not accounted for in either kink blocking or step pinning mechanisms. Evidence of tautomer occlusion can be inferred from TEM images of NH$_4$HU crystals prepared at pH 7, showing irregularities in the crystal structure (Fig. 4d) and SAED patterns (Fig. 4e) with diffuse scattering. High magnification TEM images show numerous dislocations (Fig. 4f) that are responsible for the natural bending (Fig. 1h) and branching of dumbbell-shaped crystals (Fig. 1k) at low and high urate

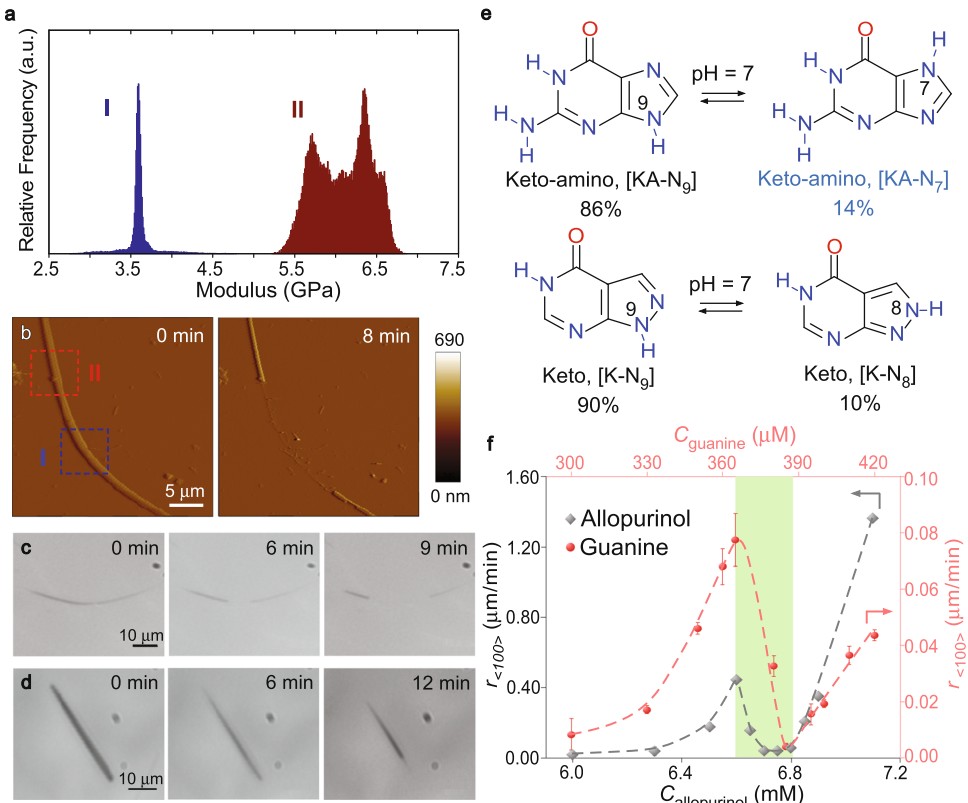

**Fig. 5 | Impact of minor tautomers on crystal properties. a** Elastic modulus of NH₄HU crystals prepared at pH 7 using contact resonance AFM. Data compare regions marked in panel **b** with relatively high (I) and low (II) defect density. **b** Time-elapsed AFM images of an NH₄HU crystal prepared at pH 7 during in situ dissolution in DI water. **c** Snapshots from Supplementary Movie 7 showing the dissolution of NH₄HU crystals prepared at pH 7 in a microfluidics device. Images were collected over 18 min in DI water. **d** Snapshots from Supplementary Movie 8 showing the dissolution of an NH₄HU crystal prepared at pH 11 in a microfluidics device. Images were collected over 36 min. **e** Chemical equilibrium between guanine (top) and allopurinol (bottom) at pH 7. Percentages of tautomers are based on first-principles (CBS-QB3) calculations at 25 °C (see Supplementary Figs. 13, 14). **f** Microfluidics measurements of macroscopic growth rates at pH 7 along the <100> direction for guanine (red) and the <100> direction for allopurinol (grey) as a function of their respective solute concentrations. Data were the average measurements of more than 15 crystals and error bars span two standard deviations. Dashed lines are interpolated to serve as a guide to the eye. The green-shaded region corresponds to growth cessation.

concentrations, respectively. In contrast, TEM images and SAED patterns of NH₄HU crystals prepared at pH 11 contain few defects (Supplementary Fig. 6g) owing to trace quantities of the minor tautomer, which results in crystal growth without noticeable inhibition.

Comparison of the electrostatic potential of major and minor tautomers (Fig. 4g) reveals a delocalization of negative charge that leaves one-half of the molecule with similar molecular recognition (i.e., unhindered binding to crystal surfaces) and the positioning of negative charge that limits [DKE-N₉]⁻ interaction with NH₄HU crystal surfaces. Notably, the inhibitor cannot bind to all kink sites equivalently owing to steric hindrance when the negatively-charged segment is oriented directly into the crystal surface, thus breaking the hydrogen-bonding of the crystal structure. The unit cell of NH₄HU (Fig. 4h, dashed box) is comprised of four distinct layers in the [101] direction, labeled L1 to L4, where the orientation of urate molecules and interstitial hydrogen-bonding between layers (Fig. 4h, dashed lines) are distinct. As illustrated in Fig. 4i, this suggests the minor tautomer [DKE-N₉]⁻ can only bind to every other layer when the negatively-charged group is oriented away from the step site. This binding motif is analogous to that observed for L-cystine crystals[46] where symmetric and asymmetric modifiers were shown to exhibit distinct binding modes. Notably, the inhibitor L-cystine methyl ester, which is an asymmetrical modification of solute L-cystine, resulted in tapered crystals. Here we observe a similar effect for NH₄HU crystallization where the asymmetry imposed by negative charges in the two tautomers yields tapered crystals (Fig. 4j) with a tapering angle φ that varies between 72 and 90°,

consistent with conceptual models of vicinal surfaces with high to low inhibitor coverage on NH₄HU crystals (Supplementary Fig. 12f). This also agrees with AFM measurements of 2D island and layer protrusion advancement at angles $\theta_1 = 81 \pm 2°$ and $\theta_2 = 86 \pm 1°$ (Fig. 2b), which have average values that lie within the range of macroscopic tapering ($\phi = 86 \pm 1°$) measured from SEM images (see also Supplementary Fig. 12c).

## Impact of minor tautomers on crystal properties
The occlusion of minor tautomer as defects impacts the physical properties of NH₄HU crystals prepared at pH 7. AFM nanomechanical characterization reveals that crystal segments containing a high percentage of defects (Fig. 5a, region I) have a lower elastic modulus compared to segments containing fewer defects (Fig. 5a, region II). These differences in mechanical properties impact the relative rates of crystal dissolution. For instance, in situ AFM (Fig. 5b) and microfluidic (Fig. 5c) dissolution measurements in deionized (DI) water reveal inhomogeneous dissolution where segments with high defect density dissolve faster. Similar measurements with less defective crystals prepared at pH 11 (Fig. 5d) reveal homogeneous dissolution. The ability to selectively control physicomechanical properties by defect engineering has practical implications for applications such as pharmaceuticals, where dissolution is crucial for oral bioavailability. To assess the broader applicability of our findings, we examined two additional tautomeric compounds (Fig. 5e). The first is allopurinol, an anti-gout drug[15,47] exhibiting major and minor tautomers at neutral

conditions (Supplementary Fig. 14). Microfluidic measurements of allopurinol crystallization show a trend of growth cessation (Fig. 5f) that is similar to that of urate (Fig. 1g). The second molecule tested is guanine, which is a biogenic crystal widely distributed in animal coloration and visual systems[13,48,49]. Guanine exists in two tautomeric forms (Fig. 5e) and has an identical region of growth cessation (Fig. 5f) owing to the appreciable concentration of minor tautomer (>10%), which was confirmed by DFT calculations (Supplementary Fig. 13).

## Discussion

The three examples reported in this study support a more generalized phenomenon of self-inhibition among tautomeric crystals when growth conditions lead to an appreciable concentration of minor tautomer(s) that function as native inhibitors of crystallization. This mode of action is unique among modifiers reported in literature where the molecular structure and/or composition of an inhibitor typically differs from the solute. Examples include constitutional isomers[50], stereoisomers[1], tailor-made inhibitors or imposters[8,46], and (macro) molecules with similar functional moieties as the solute[51]. Here we show that the presence of minor tautomer during crystallization leads to growth cessation under supersaturated conditions. These conditions can be used to tailor crystal properties, such as the rate of dissolution, which is critical for pharmaceuticals. Among the top 200 drugs, there are 33 (including allopurinol) containing tautomers that are prescribed for HIV, epilepsy, COVID-19, schizophrenia, cancer (e.g., skin, lung, and pancreatic), and other diseases that impact millions of people worldwide[10,12,15]. Minor tautomers also effect other crystal properties, such as elastic moduli that is seemingly related to natural bending. This phenomenon reported here for urate is less frequently reported[52,53] in comparison to more common examples where malleable crystals bend only under an applied force[54,55]. The broader relevance of tautomers in crystallization extends to biological systems where the role of natural inhibitors (minor isomers) is elusive. Here we focused on guanine crystals as a representative example owing to their important functional roles in many animals, where these unique properties have also inspired biomimetic engineering of photo/temperature-responsive materials[17–19,56]. It remains to be determined how the control of major/minor tautomers in natural and synthetic crystals impacts their electronic, magnetic, and/or optical properties.

## Methods

### Bulk crystallization assays

Ammonium urate (NH$_4$HU) crystals were synthesized by dissolving appropriate amounts of uric acid sodium salt powder in 20 mL glass scintillation vials containing an aqueous solution of 100 mM NH$_4$Cl and DI water under rapid agitation at 80 °C. The final solution with a total volume of 15 mL was prepared with equimolar NaHU$_{(aq)}$ and NH$_4$Cl$_{(aq)}$ ranging from 4 to 14 mM. The solution pH (ca. 7) was measured using an Orion Dual Star pH benchtop meter with a ROSS Ultra electrode (8102BNUWP). The sample vials were left undisturbed at 21 ± 1 °C for 24 h to allow crystallization, which was not affected by the presence of NaCl (Supplementary Fig. 15). The crystallization of NH$_4$HU was also carried out at pH 7 and 11 using the same concentration range without NaCl. In these experiments, the NH$_4$HU solution was prepared by dissolving appropriate amounts of anhydrous uric acid (UA) powder in 100 mL glass bottles containing DI water and adjusting the pH with 28 wt% ammonium hydroxide (NH$_3$·H$_2$O). The crystalline product (in solution) was observed by optical microscopy using a Leica DMi8 instrument. Crystals were also extracted at a periodic time and dried in air overnight. The crystalline phase was analyzed with a Siemens D5000 X-ray diffractometer (XRD) using a Cu Kα source (40 kV, 30 mA) and was confirmed by a reference pattern from the solved crystal structure (vide infra). Ex situ microscopy measurements were performed using an FEI 235 dual-beam focused ion beam scanning electron microscopy (SEM). The samples for SEM were coated with ca.

20 nm gold to reduce the effects of electron beam charging. The solubility of NH$_4$HU crystals at high pH was determined by dispersing purified NH$_4$HU powder (5–15 mg) in 20 mL vials containing ca. 15 mL of DI water and adjusting the alkalinity (pH 11.0 ± 0.1) with 28% w/w ammonium hydroxide. The vials were sealed with plastic caps and agitated for 24 h at 21 ± 1 °C to accelerate the time to reach solid-liquid phase equilibrium. The identical incubation procedure was repeated for three temperatures (25, 30, and 35 °C). The concentration was monitored over time (5–48 h) to confirm the system reached solid-liquid phase equilibrium. This was accomplished by taking small aliquots of each solution and measuring the UV-Vis absorbance with a Beckman Coulter DU 800 spectrophotometer in a 4 mL quartz cuvette with a 1 cm path length. The concentration was determined using an extinction coefficient of 1.3 ± 0.1 mM$^{-1}$ cm$^{-1}$ at a wavelength of 292 nm. Three independent measurements were performed for each temperature and only the average solubility is reported (Supplementary Fig. 4c).

### Preparation of NH$_4$HU crystal seeds

Ammonium urate crystal seeds were prepared in two steps: the spontaneous crystallization from a concentration of 7.5 mM NH$_4$HU at pH 11; and the transfer of this suspension into 400 mL freshly prepared 5.5 mM NH$_4$HU growth solution (pH 11) in a 500 mL glass bottle. The NH$_4$HU solution was prepared by dissolving appropriate amounts of uric acid anhydrous powder in DI water and adjusting the solution to pH 10.2 with 28 wt% ammonium hydroxide (NH$_3$·H$_2$O) under rapid agitation at 72 °C. The resultant solution (pH 11) was left at 21 ± 1 °C for 48 h to allow crystallization, and the obtained crystals were transferred into a 5.5 mM NH$_4$HU growth solution for a 7-day incubation period to produce needle-like crystals of length 10–50 μm and width <1 μm. To increase the width of NH$_4$HU crystals, the product after 7 days of growth was recovered by filtration and the solids were placed into another 5.5 mM NH$_4$HU growth solution for further incubation. The two-step seeded growth procedure produces crystals with a width of 1–2 μm.

### Structure determination of NH$_4$HU crystals

Crystal seeds prepared above were too small for structural determination by single-crystal X-ray diffraction; therefore, we obtained structural information from TEM using 3D ED. The NH$_4$HU crystals were crushed with a mortar and pestle, and then dispersed in pure ethanol. After 5 min of ultrasonication, a droplet of the suspension was transferred onto a copper grid, which was then loaded on a Fischione model 2020 tomography holder. The 3D ED measurements were performed on a Themis Z aberration-corrected electron microscope operated at 300 kV with a 4096 × 4096 Gatan Oneview CMOS camera (15 μm × 15 μm pixel size). The microscope is equipped with a monochromator, which was used for adjusting the beam current drawn from the Schottky field-emission electron source. The focus of the monochromator was adjusted to around 200. The microscope was operated in STEM mode during data collection. A quasi-parallel beam was obtained by adjusting the current of the C3 lens until a sharp central spot was observed on the fluorescence screen. The spot size was 10 and the C2 aperture was adjusted to 50 μm to obtain a quasi-parallel probe with a size of around 200 nm in diameter. The camera length was kept at 185 cm and the highest resolution shell was around 0.85 Å. Additional details of 3D ED data collection and structure determination are provided in Supplementary Section 1.

Powder XRD analysis was used to verify and confirm the structure determination from 3D ED. XRD measurements of NH$_4$HU crystals were performed at room temperature using a Bruker D8 Advance diffractometer. NH$_4$HU crystals were loaded into a 0.7 mm borosilicate glass capillary and mounted on the diffractometer operating in a Debye–Scherrer transmission geometry, equipped with Johansson monochromator using Cu Kα$_1$ radiation (λ = 1.5406 Å) and LynxEye

detector, operating at 40 kV and 50 mA. Additional details of crystal structure analysis are provided in Supplementary Section 1.3.

## Microfluidic assays

We employed a microfluidic platform described in previous studies[35,57,58] for in situ analysis of crystal growth kinetics. We tailored the microfluidic device (poly(dimethylsiloxane) on glass) into a cuboid house (ca. 0.5–0.8 mL) where NH$_4$HU seed crystals landing on the glass substrate were placed in the center (Supplementary Fig. 3c). This system was able to monitor the growth of NH$_4$HU crystals under continuous supply of fresh growth solution using an inverted optical microscope (Leica DMi8 instrument). Growth solutions prepared with different concentrations (5–12 mM NaHU/NH$_4$Cl or NH$_4$HU) and alkalinity (pH 7 or 11) were used in the microfluidics studies. For NH$_4$HU aqueous solution, an appropriate volume of ammonium hydroxide was added to a suspension of UA powder with pH adjustment. The growth solution was then delivered to the device by a dual syringe pump (CHEMYX Fusion 4000) at a rate of 6 mL h$^{-1}$ for at least 120 min. The concentration of the growth solution was determined using an extinction coefficient of 1.2 ± 0.1 and 1.3 ± 0.1 mM$^{-1}$ cm$^{-1}$ at wavelengths of 290 and 292 nm for neutral and high pH, respectively[32]. Time-resolved optical micrographs of crystals were analyzed using Image J (NIH)[59] for quantifying incremental changes in crystal dimension along [010] and [101] directions. The growth rate was measured by linear regression of crystal size over time (e.g., Supplementary Fig. 3g, h). The average growth rates were reported from measurements of 10 to 15 crystals. The flow rate of the growth solution was adjusted to a sufficiently high value to circumvent the influence of diffusion and ensure growth rate measurements were performed in the kinetically-controlled regime (Supplementary Fig. 4a).

Samples for dissolution were prepared on microscope glass slides (Corning) cut to a 9 × 9 mm$^2$ size. Glass slides were coated with a thin layer of curable epoxy (Loctite, China) and partially cured in an oven for 2 h at 60 °C prior to sample preparation. Crystals suspended in a saturated solution were mounted on the slide by placing a droplet and allowing 1 h for crystals to settle on the epoxy. The surface was washed with a supersaturated solution (4 mM NH$_4$HU, pH 7) for 16 h prior to drying at ambient conditions. Microfluidics experiments were performed in DI water (pH 7) and a flow rate of 6 mL h$^{-1}$ using the same procedure described above.

## Analysis of crystal defects

High-resolution TEM observations were performed on the TEAM I electron microscope (NCEM-Molecular Foundry Lawrence Berkeley National Laboratory) by dose fractionation with low-dose rates[60]. In brief, a microscope is operated using a Nelsonian illumination[61], a Cc corrector[62], and a K2 camera[63]. The illumination creates a pencil-like, highly-coherent beam of $\triangle E \lesssim 100 \, meV$ that matches the field of view (FoV) of the K2 camera (~10$^7$ Å$^2$ at high resolution with an apparent pixel size of 0.3 × 0.3 Å$^2$). This approach allows for the detection of single electron scattering events with dose rates that suppress the degradation of structure information with dose accumulation that is common to radiation-sensitive matter. It also maintains high standards for aberration corrections[64]. See Supplementary Section 1.5 for additional details.

## Atomic force microscopy (AFM)

In situ AFM measurements were performed on a Digital Instruments Multimode Nanoscope IV (Santa Barbara, CA) to examine topographical images of NH$_4$HU crystals and capture the dynamics of (20$\bar{2}$) surface growth in real-time. NH$_4$HU crystal seeds (vide supra) were filtered, immobilized on a glass slide (1 × 1 cm$^2$), and mounted on an AFM specimen disk (Ted Pella) by pressing the glass slide to the transfer crystals. AFM images were collected in tapping mode using Olympus BL-AC40TS probes (silicon nitride, Cr/Au coated 5/30,

0.09 N/m spring constant) with a tapping frequency of 30 kHz in a fluid. Image sizes ranged from 300 nm to 1.5 μm with collected using scan rates of 1 to 3 Hz and 256 scan lines at angles depending on the orientation of the monitored crystal[6,38]. The temperature in the fluid cell (29 °C) reached a steady state value after ca. 20 min imaging, which was confirmed by an AFM kymograph collected with disabled scanning along the y-axis to confirm the solubility.

Growth solutions at different alkalinity (pH 7 and 11) and urate concentrations (2–12 mM) were freshly prepared within 1 h of each measurement. The concentration of growth solutions before use was also measured by UV-Vis spectroscopy with an extinction coefficient of 1.2 ± 0.1 and 1.3 ± 0.1 mM$^{-1}$ cm$^{-1}$ at wavelengths of 290 and 292 nm for neutral and high pH, respectively[32]. The growth solution was continuously delivered to the AFM liquid cell using a dual syringe pump (CHEMYX, Fusion 200). After loading, the solution was left standing for 10–20 min to thermally equilibrate. The crystal edges were first identified in a low-magnification image to determine the orientation and the crystallographic directions. The scan direction was set parallel to the [010] crystallographic direction and AFM images were collected for 3–5 h. For studies of crystal branching at high urate concentration in a neutral solution, we first introduced a slightly supersaturated growth solution (4.0 mM urate) to the AFM liquid cell and allowed the system to reach equilibrium prior to switching to a desired urate concentration. The evolution of NH$_4$HU crystal surfaces was characterized by the velocity of advancing steps $v$ and the rate of 2D nucleation of new crystal layers J$_{2D}$, as described in Supplementary Section 1.6.

Contact resonance (CR) AFM was conducted with an Asylum MFP-3D Infinity AFM (Asylum Research, an Oxford Instrument Company, CA) in an ambient environment. Prior to any measurements, the deflection sensitivity of the AFM cantilever (Multi75G, Budget Sensors) was calibrated by force curves on a silicon surface (freshly cleaned by Piranha solution (volume ratio 3:1 for 98% H$_2$SO$_4$ and 35% H$_2$O$_2$))[65,66]. The spring constant $k_c$ of the cantilever was calibrated by fitting the first free resonant peak to equations of a simple harmonic oscillator to measure the power spectral density of the thermal noise fluctuations in air[67,68]. For CR-AFM, the ultrasonic actuation was achieved by gluing the glass substrate (with biomineral crystals on the top surface) to an ultrasonic transducer with a broadband resonance of 2.25 MHz (V133-RM, Olympus NDT)[65,66,69]. The built-in dual actuation resonance tracking approach of the Asylum MFP-3D AFM was used to track the CR frequency while scanning the sample surface in the straight and bent regions and the total applied force $F$ during the scanning was recorded[65]. The AFM cantilever was modeled as an Euler–Bernoulli beam oscillating with a mechanical constraint at the tip position (see Supplementary Section 1 for details) to extract the tip-sample contact stiffness $k^*$[69–71]. The latter can be converted to the reduced modulus of the tip-sample contact by contact mechanics models. Here, we used the most widely used Hertzian contact model, which approximates the AFM tip-sample contact as a spherical indenter with radius $R$ contacting a flat surface with force $F$[69–71]:

$$k^* = \sqrt[3]{6FRE^{*2}} \tag{1}$$

where $E^*$ is the reduced modulus and is related to the material's elastic property by

$$\frac{1}{E^*} = \frac{1-\upsilon_s^2}{E_s} + \frac{1-\upsilon_t^2}{E_t} \tag{2}$$

where $E_s$ and $E_t$ are the Young's moduli, and $\upsilon_s$ and $\upsilon_t$ are the Poisson's ratios of the sample and the tip, respectively. The tip position on the cantilever and the tip radius $R$ were calibrated by CR-AFM measurements on a sample with known stiffness (see Supplementary Section 1.7 for more details).

AFM measurements for dissolution were performed in contact mode using a Cypher ES (Asylum Research, Santa Barbara, CA) and silicon nitride probes with a gold reflex coating. Samples were prepared on a glass slide and were attached to specimen disks (Ted Pella) covered with a thin layer of epoxy (Loctite, China). In situ experiments were conducted at ambient temperature by flowing DI water (pH 7) at 6 mL h$^{-1}$ into the liquid cell (ES-Cell-Gas) with a scan rate of 1 Hz at 256 lines per scan.

## First-principles calculations

All calculations were performed with the software package Gaussian 09[72]. To obtain accurate geometries and total energies of all possible monovalent and divalent forms of urate tautomers as well as guanine and allopurinol tautomers, we used the complete basis-set, CBS-QB3[73], level of theory. Solvation effects were considered by applying the SMD[74] solvation model with the solvent being water ($H_2O$). Boltzmann distribution analysis was applied to calculate the population distribution of each possible tautomer at 25 °C using the CBS-QB3 calculated free energies. Density functional theory (DFT) calculations at the M06-2X[75,76] level of theory with 6-311 + G(d,p) basis set were performed to address the temperature dependence on the urate tautomer population and its effect on the absorption spectra, as well as the kinetics of tautomeric interconversions. The CBS-QB3 and the M06-2X calculations show an excellent agreement in the free energy ranking of the tautomers at room temperature (Supplementary Fig. 19), yielding very similar Boltzmann distributions at room temperature. Vibrational frequency calculations were performed to confirm that each stationary point is either a minimum or a saddle point. Intrinsic reaction coordinate (IRC) calculations[77] were used to confirm the path connection between the reactant, product, and transition state in tautomer interconversion. Time-dependent DFT was applied at the M06-2X/6-311 + G(d,p) level of theory (same level of theory for the temperature-dependent free energy calculations) to calculate the UV-Vis spectra of monoanionic urate tautomers. Details of the tautomer interconversion reaction calculations are provided in Supplementary Section 1.8.

## Spectroscopic verification of urate tautomers

To verify and identify the presence of urate tautomers at neutral solutions, UV-Vis, infrared (IR), and NMR spectroscopies were employed. UV-Vis spectra were recorded on a Beckman Coulter DU 800 spectrophotometer equipped with an external water circulator (Grant LTD6, ±0.1 °C, United Kingdom) for temperature control. Solutions of NaHU and NH₄HU with a concentration of 1.0 mM were prepared by dissolving known amounts of NaHU in DI water or in equimolar NH₄Cl aqueous solution and DI water at 80 °C. The clear solution of 1.0 mM NaHU was then diluted 20 times to obtain a 0.05 mM solution. These samples have a neutral pH (with values ranging between 7.0 and 7.7). A series of solutions with pH varying from 4.6 (uric acid aqueous solution) to 11.5 with a concentration of 1.0 mM urate were prepared by adjustments with either 1.0 M NaOH$_{(aq)}$ or 1.0 M HCl$_{(aq)}$. A stock solution of 1.0 M HCl$_{(aq)}$ was prepared by adding 8.3 mL of 37% HCl in 100 mL DI water. The UV-Vis spectra of 1.0 mM NaHU, 1.0 mM NH₄HU, and 0.05 mM NaHU were collected at different temperatures, beginning at 10 °C and increasing to 90 °C, and then decreasing from 90 to 30 °C in a 4 mL quartz cuvette of 1 cm path length over a spectral range from 200–400 nm and a scan speed of 120 nm/min (with a wavelength interval of 0.1 nm). The pH of 1.0 mM NaHU at increasing temperatures (21 to 90 °C) was nearly constant (i.e., varying between 7.2 and 6.9).

Fourier transform infrared (FTIR) spectra were recorded on a Thermo Scientific Nicolet 6700 instrument equipped with an Everest Diamond ATR Accessory for solid samples and a Liquid Jacketed Demountable Transmission Liquid Cell (PIKE, USA) for liquid samples. The jacketed transmission cell controlled the temperature of liquids by an external water circulator. A solution of 18 mM sodium urate in $D_2O$ (instead of $H_2O$) was prepared to circumvent the significant interferences of broad peaks at 1634 and 3300 cm$^{-1}$ from the $H_2O$ background. Solution IR spectra were collected at variable temperatures ranging from 10 to 80 °C over a spectral range from 800–4000 cm$^{-1}$ at a resolution of 2 cm$^{-1}$. Each spectrum was an average of 128 scans. The solution temperature was controlled by an external water circulator (Thermo Scientific NESLAB RTE-7) with an accuracy of ±0.1 °C. The solution pH values at different temperatures were also recorded to rule out the influence of pH changes with increasing temperature on urate speciation. Over the range of tested temperatures (22 to 80 °C), the solution pH is approximately constant (i.e. ranging from 7.6 to 7.2); thus, the population of urate monovalent species is >99.0% (Supplementary Fig. 20). For solid samples, IR spectra were collected over a spectral range of 800–4000 cm$^{-1}$ with the average of 64 scans at a resolution of 2 cm$^{-1}$.

NMR pH titration experiments were carried out on a 600 MHz JEOL spectrometer equipped with a 5-mm liquids BB probe. NMR samples of 10 mM urate-2-$^{13}$C,1,3,7-$^{15}$N₃ labeled solution were prepared by dissolving appropriate amounts of uric acid-2-$^{13}$C,1,3,7-$^{15}$N₃ crystal powders in 95% (v/v) $H_2O/D_2O$ containing 0.1% DSS and adjusting the pH by the addition of either 1.0 M HCl$_{(aq)}$ or 1.0 M NaOH$_{(aq)}$ as described above. The $^{13}$C-NMR spectra were acquired with 2048 scans at a free induction decay (FID) resolution of 0.72 Hz; and the $^{15}$N-NMR spectra were acquired with 1024 scans at an FID resolution of 0.46 Hz. For variable-temperature NMR experiments, $^{13}$C-NMR spectra were collected on an 800 MHz Bruker Avance-III spectrometer equipped with a 5-mm QCI Z-gradient cryo-probe. The temperature was controlled (BVT-3000) with an accuracy of 0.2 °C. The $^{13}$C-NMR spectra of the 10 mM urate solution (pH 7.6) were acquired by varying the temperatures from 5 to 70 °C with 128 scans at an FID resolution of 0.49 Hz. The $^{13}$C chemical shifts were determined relative to an internal standard (3-(trimethylsilyl)-1-propane-sulfonic acid sodium salt).

## Data availability

Data generated in this study are provided in the Supplementary Information as a Source Data file. The 3D ED and Rietveld refinement coordinates for structures reported in this study have been deposited at the Cambridge Crystallographic Data Centre (CCDC), under deposition numbers 2110687 and 2121918, respectively. These data can be obtained free of charge from The Cambridge Crystallographic Data Centre via www.ccdc.cam.ac.uk/data_request/cif. Source data are provided with this paper.

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

## Acknowledgements

J.D.R. acknowledges financial support from the Office of Navy Research (Grant Nos. N00014-21-1-2173 and N00014-20-1-2083) and The Welch Foundation (Grant No. E-1794). G.M. acknowledges financial support from the National Science Foundation (NSF, CBET-CAREER program) under Grant No. 1652694 and computational support from the University of Pittsburgh Center for Research Computing. A.J.F. would like to thank the EPSRC Future Continuous Manufacturing and Advanced Crystallisation Research Hub (Grant No. EP/P006965/1), ARTICULAR: ARtificial inTelligence for Integrated ICT-enabled pharmaceuticaL mAnufactuRing (Grant No. EP/R032858/1), UKRPIF (UK Research Partnership Fund) award from the Higher Education Funding Council for England (HEFCE) (Grant No. HH13054) for funding this work. X.Z. received financial support from the Swedish research council (Grant Nos. VR 2017-04321 and VR 2019-00815) and the Knut & Alice Wallenberg Foundation (Grant Nos. 2012.0112 and 2018.0237). Electron microscopy is supported by the Molecular Foundry, which is supported by the Office of Science, the Office of Basic Energy Sciences, and the US Department of Energy under Grant No. DE-AC02-05CH11231 (proposal 6225). Q.T. acknowledges the support by the startup funds from the Texas A&M Engineering Experiment Station (TEES). W.T. acknowledges financial support from the National Natural Science Foundation of China (Grant No. NNSFC 22278300). We also thank Dr. Si Li for her help with dissolution measurements and Prof. Junbo Gong for providing access to laboratory facilities.

## Author contributions

J.D.R. supervised the project. W.T. performed $NH_4HU$ bulk, microfluidics, and in situ AFM crystallization assays, FTIR and UV-Vis measurements, and prepared samples for additional characterization; X.G. assisted with bulk crystallization assays and sample preparation; W.T. and J.D.R. analyzed and discussed the data. G.M., C.A.M.-R., and S.H. performed first-principles and DFT calculations; T.Y. and X.Z. performed data collection and ab initio structure determination by 3D ED; V.K.S. and A.J.F. collected and analyzed powder XRD for structure refinement; H.M. performed liquid NMR measurements; Q.T. performed AFM nanomechanical characterization; H.A.C., C.K., and F.C.R.H. performed high-resolution TEM measurements. W.T. and X.K. performed the analyses of allopurinol and guanine crystallization; V.P.C. performed ammonium urate dissolution experiments; all authors contributed to the writing and editing of the manuscript.

## Competing interests

The authors declare no competing interests.
