## [Peer Review File · Nature Communications]

Tautomerism Unveils a Self-inhibition Mechanism of CrystallizationEditorial Note: This manuscript has been previously reviewed at another journal that is not operating a transparent peer review scheme. This document only contains reviewer comments and rebuttal letters for versions considered at Nature Communications.

Reviewers' Comments:

Reviewer #1:

Remarks to the Author:

The authors have performed a very detailed and technical study, combining experimental and computational procedures to investigate the influence of tautomers to inhibit the crystallization of ammonium urate from solution. They also provided the first detailed structural determination of ammonium urate. The results suggest that the presence of minor urate tautomers can lead to a dead-zone in the crystal growth of ammonium urate. This self-inhibition mechanism is also proposed for other two model systems, allopurinol and guanine.

Although it is well known that impurities can, even at low concentrations, affect the crystal growth rate of specific crystal faces, the observation that tautomers can promote self-inhibition seems new and interesting. From a technical point of view, the results seem quite solid and I cannot find any flaws, especially for what concerns the procedures applied. However, some of them, such as AFM and theoretical calculations, are out of my expertise.

The authors have addressed most of the previous reviewers' comments and did a very good job. However, I have some comments that should be taken into account before publication.

The manuscript still lacks in readability and sometime is too technical. For example: "Three-dimensional electron diffraction (3D ED) was performed at low electron dosage by continuously tilting the crystal and simultaneously collecting a series of electron diffraction frames at different tilt angles. The 3D reciprocal lattice of NH₄HU crystal reconstructed from the 3D ED data using the rotation electron diffraction processing program REDp33 (Fig. 1d,e, Supplementary Fig. 1a-c, Supplementary Movie 1) shows that the crystal has a monoclinic space group with $a=21.385(4)$ Å, $b=3.5300(7)$ Å, $c=20.080(4)$ Å, $\beta=114.10(3)^\circ$."

In my opinion, the information about the tilting and the software used to reconstruct the 3D reciprocal lattice, though represent important steps of the experimental procedure, are not really helping to make this part of the manuscript simple and easy to read. I guess this information could easily go to methods and this part be written more concisely. There are several other examples in the manuscript similar to that described above. The authors should consider to improve the readability of the manuscript.

Page 2, line 74. "We first prepared NH₄HU crystals in an alkaline ammonium urate growth solution (pH 11), outside the range of physiological pH as a reference condition (control)."

I do not understand why the authors mention "physiological pH" here. I agree pH 11 is outside physiological pH but I would define it as basic pH.

Microfluidics measurements. "The rate of growth, $r_{\langle 010 \rangle}$, was measured in an alkaline medium (pH 11) as a function of urate concentration, C_{urate} , using a continuous flow rate of supersaturated growth solution (see Methods)."

It is not clear here what C_{urate} means. Is it C_{urate} the total concentration of urate or just the monovalent HU⁻ (which is only the 17% of the total concentration)?

Revisions for Manuscript NCOMMS-22-46695-T

Response to Reviewer 1

We thank the reviewer for their positive comments, which include: “The authors have performed a very detailed and technical study, combining experimental and computational procedures to investigate the influence of tautomers to inhibit the crystallization of ammonium urate from solution. They also provided the first detailed structural determination of ammonium urate. The results suggest that the presence of minor urate tautomers can lead to a dead-zone in the crystal growth of ammonium urate. This self-inhibition mechanism is also proposed for other two model systems, allopurinol and guanine. Although it is well known that impurities can, even at low concentrations, affect the crystal growth rate of specific crystal faces, the observation that tautomers can promote self-inhibition seems new and interesting. From a technical point of view, the results seem quite solid and I cannot find any flaws, especially for what concerns the procedures applied...The authors have addressed most of the previous reviewers’ comments and did a very good job.” We have addressed the three additional comments from the last round of revision. Our responses and details of each action item are provided below.

Comment 1: The manuscript still lacks in readability and sometime is too technical. For example: “Three-dimensional electron diffraction (3D ED) was performed at low electron dosage by continuously tilting the crystal and simultaneously collecting a series of electron diffraction frames at different tilt angles. The 3D reciprocal lattice of NH₄HU crystal reconstructed from the 3D ED data using the rotation electron diffraction processing program REDp33 (Fig. 1d,e, Supplementary Fig. 1a-c, Supplementary Movie 1) shows that the crystal has a monoclinic space group with $a=21.385(4)$ Å, $b=3.5300(7)$ Å, $c=20.080(4)$ Å, $\beta=114.10(3)^\circ$.” In my opinion, the information about the tilting and the software used to reconstruct the 3D reciprocal lattice, though represent important steps of the experimental procedure, are not really helping to make this part of the manuscript simple and easy to read. I guess this information could easily go to methods and this part be written more concisely. There are several other examples in the manuscript similar to that described above. The authors should consider to improve the readability of the manuscript.

Response: We thank the reviewer for pointing this out to us. We have removed the nonessential technical details from the 3D ED discussion on page 2 of the manuscript and placed them in the Methods section. We have also carefully read through the entire manuscript to reduce any unnecessary technical details that are more appropriate for the experimental methods section(s).

Comment 2: Page 2, line 74. “We first prepared NH₄HU crystals in an alkaline ammonium urate growth solution (pH 11), outside the range of physiological pH as a reference condition (control).” I do not understand why the authors mention “physiological pH” here. I agree pH 11 is outside physiological pH but I would define it as basic pH.

Response: We thank the reviewer for catching this oversight. This statement is a remnant of an earlier version of the manuscript when greater emphasis was placed on pathological implications of this work. Since we have broadened the applications and impact of this work, the statement on page 2 is no longer relevant. We have edited the sentence to read as follows:

“We first prepared NH₄HU crystals in an alkaline ammonium urate growth solution (pH 11) as a reference condition (control).”

Comment 3: Microfluidics measurements. “The rate of growth, $r_{<010>}$, was measured in an alkaline medium (pH 11) as a function of urate concentration, C_{urate} , using a continuous flow rate of supersaturated growth solution (see Methods).” It is not clear here what C_{urate} means. Is it C_{urate} the total concentration of urate or just the monovalent HU^- (which is only the 17% of the total concentration)?

Response: We thank the reviewer for bringing up this point. The term C_{urate} represents the total concentration of all urate species. To improve the clarity of this statement, we have corrected the sentence as follows:

“The rate of growth, $r_{<010>}$, was measured in an alkaline medium (pH 11) as a function of total concentration of urate species, C_{urate} , using a continuous flow rate of supersaturated growth solution (see Methods).”